# Identifying Environmental Impact Factors for Sustainable Healthcare: A Scoping Review

**DOI:** 10.3390/ijerph20186747

**Published:** 2023-09-12

**Authors:** Marieke Sijm-Eeken, Monique Jaspers, Linda Peute

**Affiliations:** 1Department of Medical Informatics, Amsterdam Public Health Research Institute, Amsterdam UMC Location University of Amsterdam, Meibergdreef 9, 1105 AZ Amsterdam, The Netherlands; 2Center for Sustainable Healthcare, Amsterdam UMC, Meibergdreef 9, 1105 AZ Amsterdam, The Netherlands; 3Center for Human Factors Engineering of Health Information Technology, Amsterdam UMC, Meibergdreef 9, 1105 AZ Amsterdam, The Netherlands

**Keywords:** environmental pressures, measurement, quality management, climate, emissions, planetary health

## Abstract

The healthcare industry has a substantial impact on the environment through its use of resources, waste generation and pollution. To manage and reduce its impact, it is essential to measure the pressures of healthcare activities on the environment. However, research on factors that can support these measurement activities is unbalanced and scattered. In order to address this issue, a scoping review was conducted with the aims of (i) identifying and organizing factors that have been used to measure environmental impact in healthcare practice and (ii) analyzing the overview of impact factors in order to identify research gaps. The review identified 46 eligible articles publishing 360 impact factors from original research in PubMed and EBSCO databases. These factors related to a variety of healthcare settings, including mental healthcare, renal service, primary healthcare, hospitals and national healthcare. Environmental impacts of healthcare were characterized by a variety of factors based on three key dimensions: the healthcare setting involved, the measurement component or scope, and the type of environmental pressure. The Healthcare Environmental Impact Factor (HEIF) scheme resulting from this study can be used as a tool for selecting measurable indicators to be applied in quality management and as a starting point for further research. Future studies could focus on standardizing impact factors to allow for cross-organization comparisons and on expanding the HEIF scheme by addressing gaps.

## 1. Introduction

The global healthcare industry has a negative impact on the Earth’s environment, especially in developed countries [1,2,3,4]. The 2022 Lancet Countdown Report showed an increase in healthcare’s emissions to 2.7 Gt CO_2_equivalent (CO_2_eq), which accounts for 5.2% of global emissions [1]. Besides greenhouse gas (GHG) emissions, healthcare activities impact the environment by other factors that exert negative effects on human health, such as pollution of water, air and soil, waste generation, and using scarce resources such as fresh water [4,5]. These impacts on the ecological environment have effects that further damage Earth’s ecosystem, such as global warming and climate change [4]. As the mounting evidence and awareness of healthcare’s environmental footprint grows, it is becoming clear that healthcare professionals and decision-makers have an ethical responsibility to transform healthcare in favor of the environment.

A promising approach for transforming healthcare and reducing its environmental footprint is to apply methods from the field of quality management (QM). QM not only supports the integration of sustainability in decision-making, but also facilitates implementation of environmental management systems [6]. The concept of integrating environmental sustainability and QM in healthcare is not broadly accepted yet, even though it has been applied in some case studies [7,8]. Nonetheless, the International Society for Quality in Health Care (ISQua) recognizes sustainability as a key priority in healthcare [9].

Facilitating and expanding the application of QM methods to mitigate the negative impacts (or pressures) of healthcare on the environment requires insight into how to measure and quantify the environmental impact of various healthcare services. For example, in his widely adopted “14 steps” approach, Crosby recommends: “establish quality measurements for activities where they don’t exist and review them where they do” [10]. Another example is how Mortimer et al. [6] included environmental impacts in the definition of value, which is central to QM methods in the healthcare setting:(1)$$Value\;of\;Care=(\frac{Outcomes\;for\;patients\;and\;populations}{Environmental\;+\;Social\;+\;Financial\;Impacts})$$

Including environmental impacts in QM methods can only be realized if these can be quantified. Quantifying environmental impacts is not a new concept, and numerous methods have already been developed to assist with this endeavor. Three types of approaches to support this quantification are: calculation, measurement or a mixture of both [11] (p. 314). Calculations are performed by using models; examples include mass-balance approaches, GHG activity multiplication, environmental impact assessment and lifecycle assessment (LCA). Applying these models requires specific expertise and takes time, which makes them less suitable for usage in QM projects or by teams in healthcare settings. These teams typically do not have the knowledge and skills for applying advanced environmental models [12]. Measurement, on the other hand, entails the collection of primary data, either continuously or periodically, which is an already common activity for healthcare professionals involved in QM. The aforementioned underscores the significance of distinguishing between measurement and calculation. From a practical standpoint, measurement offers significant benefits.

Although much has been written on how environmental impacts can be measured and quantified for different healthcare services, the current body of research is scattered and unbalanced. Different types of environmental impact measures have been used to quantify healthcare settings in different regions, for systems with a different nature and for different goals. Even when focusing on a single type of impact such as greenhouse gas (GHG) emissions, researchers Schmidt and Bohnet-Joschko highlighted the need for standardization of measurements [13]. As a result of the lack of insight into how environmental impact in healthcare can be measured, applying environmental measurements in quality improvement initiatives is difficult. To mitigate this problem, insight into existing practices on measuring environmental impact factors in healthcare is needed.

The aim of this scoping review is twofold. The first aim is to identify and organize factors that have been used to measure environmental impact in healthcare practice. The second aim is to analyze the overview of impact factors to identify research gaps. We use the term “environmental impact factors” in this study to refer to the measurable units describing how activities from within the healthcare system put pressure on the Earth’s natural environment. In the context of this study, the term “environment” will refer primarily to the atmosphere, biosphere and hydrosphere. Healthcare activities in this study cover the complete healthcare sector, including pharmacies, healthcare institutions and patient homes.

The findings of this review can be used to support the inclusion of environmental impact in QM activities and improvement initiatives to reduce the environmental footprint of healthcare activities.

## 2. Materials and Methods

### 2.1. Methodological Overview

For the identification of impact factors and their characteristics, a scoping review method was conducted using a predetermined protocol in accordance with the Preferred Reporting Items for Systematic Reviews and Meta-Analyses extension for Scoping Reviews (PRISMA-ScR) [14].

### 2.2. Search Strategy and Selection Criteria

The full search strategy and selection criteria are presented in the appended research protocol (Appendix A). These criteria focus on identifying articles in which primary data are measured for defining the environmental impact of healthcare organizations or activities. The literature search was conducted on 18 October 2022 on PubMed and all EBSCOhost databases combining search terms related to three clusters of topics: healthcare, impact measures and climate/environment. The PubMed database was searched because of its focus on healthcare, and the EBSCOhost databases were searched because of their broader scope, including the environmental focus. Only English language studies were considered.

### 2.3. Data Abstraction and Analysis

Articles resulting from these searches were merged in the Rayyan.ai web-application [15], in which double entries were identified and removed. MSE and LP first screened the titles and abstracts independently for eligibility and reached agreement on differences. Next, remaining articles were assessed for relevance in full-text review by MSE and LP. Disagreements were resolved in an interactive session. Finally, raw data and text were extracted by MSE and LP using a structured spreadsheet in Microsoft Excel version 2019 for which each reviewer (MSE or LP) verified the extracted data by the other reviewer (MSE or LP).

Data extraction included generic study information (study type and design; country of research setting), characteristics of the healthcare setting (type of organization, organizational unit/department, type of healthcare activity) and information describing the environmental impact and measures (scope, measurement units, values).

Factors were then mapped to the pressures-state-response framework of the Organisation for Economic Co-operation and Development (OECD) [16] based on their scope and healthcare setting. This OECD framework differentiates between direct and indirect environmental impacts. Indirect impacts are human activities, and direct impacts are the pressures of these activities on the environment. In this study, the human activities are the healthcare activities, and the impact factors describe the direct pressures on the environment resulting from these activities. Incongruencies in data analysis between the two researchers were discussed and resolved as part of the coding process.

The resulting data abstraction form was analyzed by MSE to provide an overview of knowledge gaps by comparing the overview of factors to the environmental impacts of healthcare facilities as defined by the World Health Organization (WHO) in their framework for sustainable healthcare facilities [17].

The structured data in Excel were used as input for a synthetic analysis in which factors were categorized and organized in relation to the research aims to develop overviews for presenting the factors in the most meaningful ways. This was an iterative process in which factors were grouped and related by their type of environmental impact, healthcare setting characteristics and measurement units.

## 3. Results

### 3.1. Descriptive Analysis

In total, 1765 records were identified, with 295 duplicates. Screening of the 1470 records on relevance resulted in 1256 exclusions. Full texts of the 214 remaining records were reviewed for eligibility, resulting in 46 included articles. The literature research process is summarized in Figure 1.

The identified papers were published between 2002 and 2022 and together describe 360 unique factors. The characteristics of included articles are summarized in Table 1.

Table 2 provides a detailed overview of all included articles, their characteristics and the amount of impact factors per type identified in each of the articles.

The geographic contexts in which these factors were developed were spread across and between continents as presented in Figure 2. More than half of the articles originated from two countries: the USA and UK. There were no factors reported for the continent of Africa. The work from MacNeill et al. [46] presented factors from hospitals in three countries (Canada, USA and UK).

### 3.2. Content Analysis

#### 3.2.1. Healthcare Settings, Environmental Pressures and Impact Factors

The pressure-state diagram in Figure 3 provides an overview of the healthcare activities (indirect pressures) for which factors were identified in the included articles, with the total number of impact factors (direct environmental pressures) per type (waste generation, pollutant emission, resource use) caused by these activities. Figure 3 presents the outcomes of data abstraction and coding, leading to the identification of healthcare activities for different groups of healthcare settings. In addition to the factors specifically attributed to one of the three types of environmental impacts used for coding as outlined by the OECD, a distinct category labeled as “combined pressures” was incorporated to encompass elements that entail numerous forms of environmental impacts. Details on specific factors and the corresponding healthcare activities are included in the Appendix A for waste generation, Appendix A for pollutant emissions and in Appendix A for resource use.

Factors were defined on the national level for primary healthcare, hospitals, renal services and mental healthcare. All factors could be related to seven types of activities taking place in the healthcare settings, causing direct pressures to be put on the environment.

The direct environmental pressure “waste generation” accounted for 190 impact factors (52.7%), followed by 131 factors related to the pollution of water and air (36.4%). The use of resources was described by 34 factors (9.4%), and the remaining five factors combined pollution and resource use in one metric. These five factors are not included in Figure 4, which is the result of four steps. First, the 355 unique environmental impact factors that did not combine types of pressures were grouped based on their type of environmental pressure. Each type of pressure is presented by a colored circle in Figure 4, where the color of the circle corresponds with the color of direct pressures in Figure 3 (waste generation in yellow, pollutant emission in gray and resource use in blue). The three types of environmental pressure were then further divided into sub-categories based on the component or scope (aspect of environmental impact) the impact factors addressed. These sub-categories are visualized by the colored squares in Figure 4. The measurement units used in the extracted factors related to each impact type sub-category are then listed in bold on top in each white square. Additionally, the frequency (number of factors using this measurement unit to describe this impact type sub-category) is added in blue text beneath the measurement units. Lastly, for each measurement unit within a sub-category of environmental pressure, the objects of measurement are added in the lower half of each white box in Figure 4. Figure 4 does not provide the corresponding healthcare settings with each group of factors.

Factors for resource use were divided into groups aimed at measuring the use of water, the use of energy, and factors for mixed resource use. While factors for water use were all reported per treatment, energy use was described by a multitude of units, including the energy use per square meter of a healthcare center. The measurement units “kg CO_2_/CO_2_eq per year” and “kg CO_2_/CO_2_eq per treatment” were used to describe impact factors in five and four different categories, respectively. Air pollution was in all cases defined in terms of carbon emission measures, while water pollution factors used the weight of substances besides carbon emission equivalents. Factors related to waste generation all used the weight or volume of waste, however, for various units such as per patient, per bed, per day or per year for an entire healthcare center. The measurement unit “Liters per treatment”’ was used to describe factors related to volume of waste as well as volume of water used.

#### 3.2.2. Completeness of Identified Factors

The comparison of the identified factors with the WHO guidance [17] showed that factors were identified for the WHO impact categories of water, wastes, air pollution, and greenhouse gases. The factors related to pharmaceutical residues in waste water and surface water were considered factors in the category “chemicals” as defined by WHO. No factors were identified for the categories “sanitation” and “radiation”.

## 4. Discussion

This study identified a rich set of factors describing environmental impacts based on primary data collected in different healthcare settings. Aggregating and analyzing the individual impact factors into several structured models and overviews provided several insights.

First, it showed that the environmental impact of healthcare activities is multi-dimensional. The three main dimensions that came across when aggregating and analyzing the factors in this study were: the (part of a) healthcare setting a factor relates to, the component and/or scope of measurement, and the type of environmental pressure it describes. The choices made in the studies analyzed for each of these dimensions resulted in differences in the “breadth” (mainly defined by the boundaries of the healthcare setting) and “depth” (predominantly determined by the components of measurement) of impact factors.

The geographical location could also be considered as a dimension, especially because differences in environmental statuses and public health factors between locations exist [17]. However, because most included studies originated from a few countries, analyzing the factors based on geography was not expected to add considerable value, hence geographical location was not considered as a dimension of the impact factors.

Second, a noticeable variation in the scale of measurement was observed among the impact factors identified in this study. While some factors were defined on the national or regional level, others were specific to individual procedures or individual patients. This is probably due to the lack of standardization and (international) agreements. This lack of consistency in the scale of measurement poses challenges for comparisons and benchmarking.

Besides the variety in scales of measurement, it must be noted that the definitions and measurement approaches used also varied between studies. An example is the definition and classification of different types of waste. Some authors differentiated between hazardous and non-hazardous medical waste, while others [46] further differentiated medical waste into specific subcategories such as black box waste (acutely toxic and infectious waste), cytotoxic waste and hazardous waste. Blue wrap waste (polypropylene) in some studies was considered “recyclable waste”, while in other studies blue wrap waste was not recycled but managed as municipal solid waste. For this reason, comparing values of specific impact factors between studies does not provide valuable insights.

A third obvious finding in this review is that a majority of the identified factors (190, 52.8%) related to the generation of waste. Waste reduction is an appealing subject due to its potential for cost savings, as indicated in several previous studies [63,64], and its high visibility. As opposed to other environmental impacts. waste volumes are easily observable at healthcare facilities. The effects of energy consumption and pollution of water, air and soil are typically more distal, occurring upstream at energy plants and locations where resources are harvested, or downstream in rivers where healthcare facility wastewater is released. Consequently, these impacts are less noticeable to healthcare employees and more difficult to relate to healthcare activities than waste generated.

The fourth noteworthy finding relates to the identified gaps in terms of missing factors. None of the included studies defined factors to measure environmental impact from radiation or sanitation. As radiation from healthcare services, especially in radiology settings, is known for the hazards to patients and staff [65], this is an important observation. The absence of factors for non-radioactive radiation could be explained by the lack of evidence on the impact of ambient radiation levels to human health [66]. However, this is not true for radioactive radiation, which is well known for its significant environmental and health impacts, such as after the Fukushima nuclear accident [67].

A possible explanation for the absence of factors describing sanitation is that the included studies were mainly performed in high-income countries where sanitation is mostly in place and functioning well [68]. Considering the inadequate access to sanitation in many developing countries [69], measuring availability or quality of sanitation could provide insights that can aid environmental impact assessment and improvement.

Lastly, a key observation is the absence of research defining environmental impact factors for many types of healthcare facilities. The WHO guidance used in the gap analysis is intended for all types of healthcare facilities, including hospitals, clinics and community centers of all sizes [17]. However, the research identified in this review primarily originated from hospital studies, while facilities such as those for the elderly and pharmacies were not presented.

### 4.1. Interpretation within the Context of the Wider Literature

Our finding that environmental impact factors for healthcare are multidimensional is consistent with findings from research on factors used for measuring quality of care in general [69]. That research highlighted the importance of combining factors into sets depending on the purpose, to realize sufficient coverage of the quality-of-care element to be measured. This approach is also followed by researchers to calculate the environmental footprint of the healthcare sector. Lenzen et al. [4] combined seven factors covering different categories of environmental impact to express the global healthcare sector’s footprint. Due to their targeted scope—the full healthcare sector rather than specific settings, objects or regions—they did not use multiple dimensions. Steenmeijer et al. [70] combined factors from three dimensions to calculate the environmental footprint of the Dutch healthcare sector: healthcare setting, healthcare activities and environmental impact categories. Thus, our finding of multiple dimensions being relevant for defining environmental impact factors suggests the need for considering a combination of factors for measuring environmental impact in QM initiatives in healthcare.

Our study found a wide range of variation in measurement factors for each of their dimensions. In concordance with these results, previous studies have demonstrated that variation exists in units of measurement for factors used in QM [71,72]. Common issues related to the lack of standardization include labor-intensive measurement processes, difficulties in sharing data (interoperability) and poor availability and validity of data. Further, when used for external accountability and verification, there is potential for perverse incentives putting data reliability at stake [73]. Therefore, standardizing measurement approaches and definitions can prevent these issues.

This review was unable to identify other research that reported findings similar to ours on waste being a dominant area for measuring environmental impact in healthcare. Even though healthcare waste generation was a key component used in the environmental footprint study from Steenmeijer et al. [70] for the Dutch healthcare sector, Lenzen et al. [4] did not address it in their assessment of the environmental footprint of the global healthcare industry.

### 4.2. Implications for Practice

The comprehensive methodology employed in this review, which involved utilizing multiple databases to gather environmental and medical literature, along with a structured analysis conducted by two researchers, resulted in the development of a thorough and well-organized overview of measurable impact factors. This compilation holds significant potential in enhancing healthcare practices.

#### 4.2.1. A “Menu-Card” for Selecting Environmental Impact Factors

Up until recently, there has been little guidance for healthcare professionals looking for viable approaches for measuring environmental impacts in healthcare—for instance, as part of QM initiatives. The HEIF scheme and overviews that have been established in this study can now facilitate the selection process by serving as a comprehensive tool, or “menu-card”, for identifying measurable environmental impact factors.

#### 4.2.2. Development of New Environmental Impact Factors

The insights on multiple dimensions for measuring environmental impact derived from this research could be used to develop new impact factors not yet included in the HEIF scheme. Because of the underlying evidence of the HEIF scheme predominantly being based on research from some specific hospital settings and regions, generalizability of this scheme to all healthcare settings and regions could be limited.

### 4.3. Limitations

Some limitations to this study exist that are worth noting. First, the literature search strategy was designed to minimize bias but still has some limitations. Selection bias could have led to the exclusion of relevant articles as a result of choices in search terms used in the literature review. Second, an inherent limit from applying inclusion criteria, such as the use of the English language, is that we left out studies that could have resulted in identifying more impact factors. However, given the objective of this research to focus on candidate factors for application in QM initiatives, the restriction to only include research reporting actual measurements was considered to be appropriate. As was set out in the introduction, the specific knowledge and skills required for applying models to calculate environmental impacts are not commonly available in healthcare organizations.

Another limit was the approach followed for assessing knowledge gaps. First of all, this approach aimed to assess gaps on a high aggregation level, looking for missing categories of environmental impact rather than the absence of specific factors. Even though some gaps were identified, we are confident that more factors are missing. For instance, pollution by chemicals is one of the impacts of the WHO framework that was not marked as a gap in this study, as factors describing pharmaceuticals in water were identified in one or more factors. However, it could be that pollution by chemicals is also relevant to air and soil, besides water.

Hence a more in-depth gap analysis, including multiple domains, could have resulted in the identification of more gaps. To perform this analysis, environmental expertise is needed. Further, the WHO framework used for this step focuses on human health rather than on the broader concept of planetary health. It could be argued that initiatives aiming to reduce environmental impacts should take the broader perspective and consider the impact on the planet rather than only on human health. As a result, gaps from a planetary health perspective, such as loss of biodiversity, land use or the use of scarce resources, were not identified in this study.

### 4.4. Future Research

In its current form, the HEIF scheme does not provide guidance in the selection process to obtain the best set of impact factors. Future development of the scheme could focus on this guidance, for example by considering selection criteria based on objectives and setting and obtaining a better understanding on environmental impact factor effectiveness.

Five areas for future research can contribute to expanding and enhancing the content of the HEIF scheme. A first recommendation is to address the gaps identified by comparing our overview of factors to the WHO framework, for example by adding factors describing healthcare’s impact in terms of radioactive radiation and sanitation. For sanitation, including factors to measure availability or quality of sanitation would be a good addition to the HEIF scheme. Sanitation assessment suggested by WHO [17] could include monitoring of sanitation-related deceases and overflow of sanitation infrastructure.

A second recommendation is to perform a more in-depth analysis of gaps. Each type of environmental impact could be further differentiated based on its subtypes (for example, pollution of air could be further detailed into different types of air pollution). The relevance of each of these detailed environmental impacts can be considered for measurement in the healthcare context. Based on the outcomes, the HEIF scheme developed in this study can be expanded. A scoping review to elucidate the relationships between healthcare and these detailed environmental impacts could be a good first step. A third suggestion is to also expand the breadth of the gap analysis by including environmental impact categories related to planetary health next to human health.

Another recommendation is to investigate if measuring environmental impact in healthcare settings other than the ones presented in this review, such as facilities for the elderly and pharmacies, will deliver new insights into how to measure environmental impacts.

The fifth and last recommendation related to the content of the HEIF scheme is to perform research focusing on standardization of environmental impact factors. After such a standardization process, further research could focus on the effective use of these factors in quality improvement initiatives and on increasing our understanding of how and where healthcare impacts the environment. For example, by analyzing care pathways to identify where most impact is forced on the environment (identification of “hotspots”) or by comparing and contrasting values of similar factors between healthcare settings to find best practices and accelerate improvements.

Even though we suggest expanding our HEIF scheme, we do not assume that the overview of environmental impacts caused by healthcare will ever be complete. First of all, current knowledge on how human activities impact the environment is limited [74]. Second, certain impacts cannot be measured even though instruments are continuously improving [74]. Other impact measurements require specific equipment not available in common healthcare settings. For example, measuring air pollution by fine particulate matter requires the use of an electron microscope [75]. Lastly, environmental impacts often are measured in locations distal from healthcare facilities, making it difficult or impossible to link these to healthcare. Some air pollution, for example, is measured by satellites. Like with other concepts within quality of care, such as patient satisfaction, the goal is not to have an all-inclusive insight based on measuring all elements, but to find those variables that are most relevant and to focus efforts on measuring these most essential items. Our overviews and HEIF scheme form a solid base to further move in this direction. An important next step is to identify essential variables and assess the feasibility of measurement for these variables in order to develop standardized measurements. Next, these should be validated through testing and agreed upon by healthcare bodies to become common indicators for QM in healthcare.

## 5. Conclusions

This review aimed to identify, analyze and aggregate available environmental impact factors for the healthcare industry and to identify knowledge gaps. The results show that there are many ways to measure the environmental impact of healthcare and that these can be defined by using multiple dimensions. Notwithstanding its current limitations, the study adds to our understanding of the factors to measure the environmental impact for healthcare settings. It is important to expand the resulting HEIF scheme and develop standardized definitions and measurement methods to ease data collection, improve the quality of measurements and allow for cross-organization comparisons.

As there is an urgent need to mitigate climate impact and reduce the substantial environmental impact of healthcare, the results of this study can be applied to support the measurement of environmental impacts alongside other quality of care constructs.

By doing so, we can track progress over time and make comparisons between organizations on the path to sustainable and low-carbon health systems.

## Figures and Tables

**Figure 1 ijerph-20-06747-f001:**
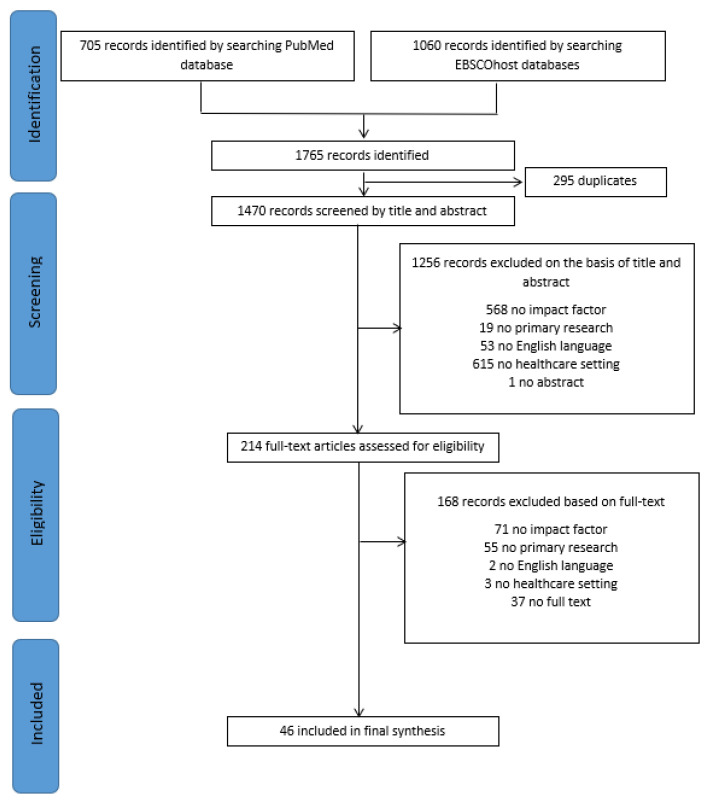
PRISMA flow diagram.

**Figure 2 ijerph-20-06747-f002:**
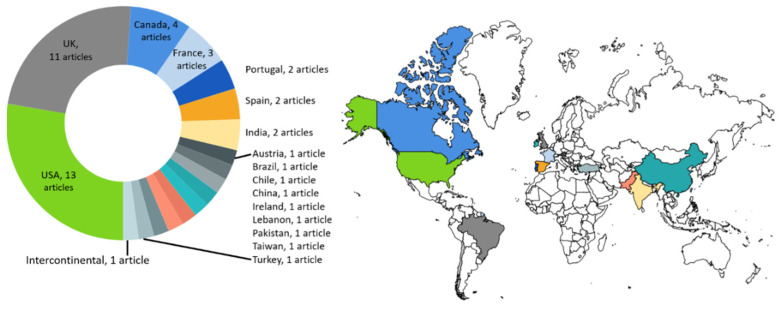
Geographical distribution of articles (country names with number of articles).

**Figure 3 ijerph-20-06747-f003:**
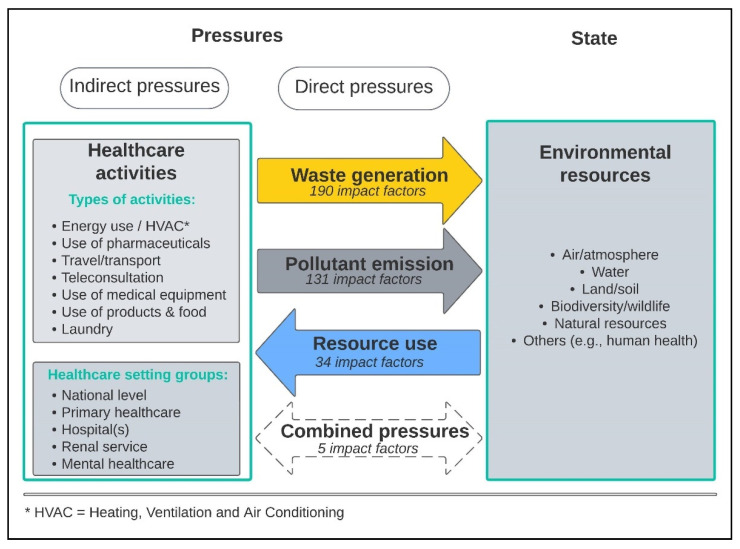
Pressure-state diagram with healthcare activities, settings and number of impact factors identified in this study (total of 360 unique factors).

**Figure 4 ijerph-20-06747-f004:**
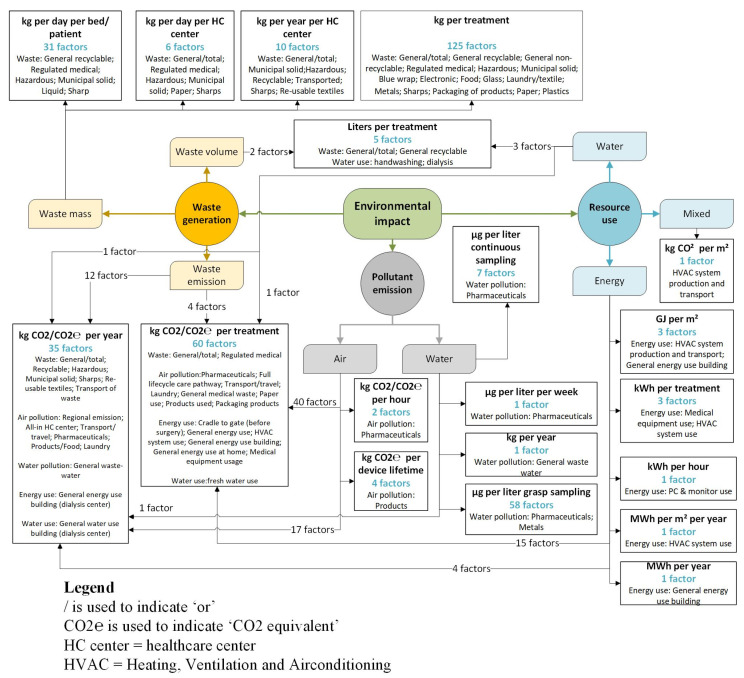
Healthcare Environmental Impact Factor Scheme (HEIF-Scheme)—environmental impact factors per impact category, grouped by measurement units. The five combined factors are not included in this scheme, which is why the total adds up to 355 rather than 360. Connecting arrows illustrate “consists of” relationships between elements of the scheme.

**Table 1 ijerph-20-06747-t001:** Descriptive summary of included articles (*n* = 46).

	% (*n*)
Geographical area (based on country of research)	
Europe	43.5 (20)
North America	34.8 (16)
Asia-Pacific	15.2 (7)
South America	4.3 (2)
Intercontinental	2.2 (1)
Healthcare setting (based on main focus of research)	
Single hospital	63.0 (29)
Multiple hospitals	19.6 (9)
National level	4.3(2)
Healthcare center	2.2 (1)
Healthcare center and home	2.2 (1)
Mental healthcare (multiple centers)	2.2 (1)
Primary healthcare (multiple centers)	2.2 (1)
Renal service	2.2 (1)
Multiple healthcare centers incl. renal services	2.2 (1)
Healthcare activities (based on main focus of research)	
Surgery ^1^	43.5 (20)
All-in setting	30.4 (14)
Radiology ^2^	6.5 (3)
Renal services	4.3 (2)
Ambulance transport coronary incidents	2.2 (1)
Emergency care	2.2 (1)
Intravitreal injection	2.2 (1)
Medication prescription (inhalers)	2.2 (1)
Teleconsultation	2.2 (1)
Use of sharps containers	2.2 (1)
Multiple departments	2.2 (1)

^1^ Multiple or single types of surgical procedures or surgery-related activities within healthcare setting incl. general surgical activities (use of laryngeal mask airway, use of anesthetics, surgical scrubbing). ^2^ Includes diagnostic and interventional radiology.

**Table 2 ijerph-20-06747-t002:** Characteristics of included articles (*n* = 46).

Article	Country and Healthcare Setting	Healthcare Activities	Environmental Impact (Pressure)—Number per Type						
			Resource Use			Pollutant Emission		Waste Generation	Mixed
			Water	Energy	Mixed	Air	Water		
Jolibois et al. (2002) [18]	France, hospital	All-in setting					1		
Gilliam et al. (2008) [19]	UK, hospital	Laparoscopic surgery				1			
Seifrtová et al. (2008) [20]	Portugal, multiple hospitals	All-in setting					4		
Somner et al. (2008) [21]	UK, multiple hospitals	Surgical scrubbing	2	1					
Lin et al. (2010) [22]	Taiwan, multiple hospitals	All-in setting					10		
Masino et al. (2010) [23]	Canada, health center	Teleconsultation		1					
Connor et al. (2010) [24]	UK, renal service	Renal services	1	4		11		4	
Zander et al. (2011) [25]	UK, hospital	Ambulance transport coronary incidents				2			
Eker and Bilgili (2011) [26]	Turkey, multiple healthcare centers incl renal services	All-in setting						30	
Riedel (2011) [27]	USA, hospital	All-in setting						2	
Eckelman et al. (2012) [3]	USA, hospital	Use of laryngeal mask airway				2			
Grimmond and Reiner (2012) [28]	USA, hospital	Use of sharps containers				2			
Power et al. (2012) [29]	USA, national	Laparoscopic surgery				3		2	
Goullé et al. (2012) [30]	France, hospital	All-in setting					3		
Tay et al. (2013) [31]	Australia, hospital	Use of anesthetics				2			
Morris et al. (2013) [32]	UK, hospital	Cataract surgery	1	1		8		1	1
Gros et al. (2013) [33]	Spain, hospital	All-in setting					16		
Southorn et al. (2013) [34]	UK, multiple hospitals	Orthopedic surgery						3	
Debois et al. (2013) [35]	USA, hospital	Coronary surgery						1	
Diwan et al. (2013) [36]	India multiple hospitals	All-in setting					14		
Lui et al. (2014) [37]	Canada, multiple hospitals	Head and neck surgery						16	
McCarthy et al. (2014) [38]	Ireland, hospital	Radiology		1					
Maamari et al. (2015) [39]	Lebanon, multiple hospitals	All-in setting						1	
Dias-Ferreira et al. (2015) [40]	Portugal, hospital	Per department						19	
Thiel et al. (2015) [41]	USA, hospital	Hysterectomy (robotic vs. human)						2	
De Sa et al. (2016) [42]	Canada, hospital	Orthopedic surgery (total knee arthroplasty)						7	
Maughan et al. (2016) [43]	UK, multiple mental healthcare centers	All-in setting		1		2			1
Chen et al. (2017) [44]	China, healthcare center and home	Peritoneal dialysis	1	2		9		2	
Jha et al. (2017) [45]	India, hospital	All-in setting					13		
MacNeill et al. (2017) [46]	Canada, UK and USA, multiple hospitals	Surgery		4		2	2	18	
Berner et al. (2017) [47]	Chile, hospital	Surgery ^1^		6		12			3
Esmaeli et al. (2018) [48]	UK, hospital	Diagnostical radiology		3					
Khan et al. (2019) [49]	Pakistan, multiple hospitals	All-in setting						4	
Wilkinson et al. (2019) [50]	UK, national	Medication prescription (inhalers)				1			
Kooner et al. (2019) [51]	Canada, hospital	Orthopedic surgery ^2^						11	
Hsu et al. (2020) [52]	USA, hospital	Emergency care						13	
De Oliveira Klein et al. (2021) [53]	Brazil, hospital	All-in setting					5		
Rammelkamp et al. (2021) [54]	USA, hospital	Surgery ^3^						44	
Garcia-Sanz-Calcedo et al. (2021) [55]	Spain, multiple primary healthcare centers	All-in setting		1	1				
Patel and Smith-Steinert (2021) [56]	USA, hospital	Surgery				1			
Cameron et al. (2021) [57]	USA, hospital	Intravitreal injection						4	
Chua et al. (2021) [58]	USA, hospital	Interventional radiology		1		2		4	
Baxter et al. (2021) [59]	UK, multiple hospitals	Wrist and radius injury surgery				1			
Wang et al. (2021) [60]	USA, hospital	Elective spinal surgery		1					
Grinberg et al. (2021) [61]	France, hospital	Coronary surgery		1		2		1	
Talibi et al. (2022) [62]	UK, hospital	Neuro surgery						1	

The numbers indicate the number of reported factors presented by the research for the corresponding type of environmental pressure. ^1^ abdominoplasty, breast augmentation, rhinoplasty ^2^ general orthopedic surgery, arthroplasty, foot and ankle, sports, trauma, upper extremity ^3^ amputation (below knee, guillotine), angiogram, biopsy of penile mass, carpal tunnel release, cataract surgery, coronary surgery (aneurysm repair, artery bypass, artery revascularization, aortic valve replacement), cystoscopy debridement of foot, deep brain stimulation, exam under anesthesia, fistulotomy, gastrocnemius repair, incision and drainage, laminectomy, laparoscopic surgery (bowel resection, gastric bypass, hernia repair, prostatectomy, sleeve gastrectomy), laryngoscopy, lobectomy, mandible repair, mouth excision, open hernia repair, orchiectomy, orthopedic arthroplasty, penile implant, radiofrequency ablation, skin lesion excision, sural nerve biopsy, transurethral resection of prostate, ureteroscopy, vasectomy.

## Data Availability

The data underlying this article are available in the article and its Appendix A.

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
