# Peer review of "Identifying Environmental Impact Factors for Sustainable Healthcare: A Scoping Review"

_ijerph, 2023, doi:10.3390/ijerph20186747_

Round 1

Reviewer 1 Report

the review article titled "Identifying environmental impact factors for sustainable healthcare: a scoping review" tried to identify the factors used for measure environmental impact in healthcare by analyze the published work. Overall, the work is very meaningful, the conclusions could be well supported by the data generated in the study. Howver, there are some issues or concerns the authors should be carefully addressed before consideration of formal publication.

Major

1. in the cureent study, seems that the authors raised a kind of new concept, i.e. environmental impact factors for sustainable healthcare. Howver, such statement is not really new, whatever the authors mentioned is quite similar to the concepts in the field of Environmental Impact Assessment in Environmental science and Environemtnal Health Saftey in environmental management, the reviewer would highly suggest the authors probe the potential relationship between the topic in the current study and the EIA an/or EHS.

2. for the searching stragtegy and selection criteria, environmental or environment were failed to be included in the the searching topics, the reviewer doubt the completeness of the returned literatures, also limitation was only set for the language, is there any limitation for literature type (review, journal article, government report, dissertation, or book chapters).

Minor

line90 briefly describe what is PRISMA-ScR, though the authors inserted a citation here.

Fig.4 as well as other places, pay attention to the format of CO2

the English is fine, only minor revision is needed.

Author Response

Thank you for the review of our manuscript and we appreciate the suggestions made. In the attached document we have provided a response to each review point. The comments are supported with a new version of our manuscript in which we have processed review comments from all reviewers.

Reviewer 2 Report

The article is well written, I only have a few drone comments

1) the preliminary analysis shows neither the purpose of the study nor the research gap

2) the hypotheses and scheme of how the study itself will look like have not been indicated

3) Recommend you rewrite the abstract, indicate the purpose of the study, approach, methodology, and the results and limitations

Author Response

Thank you for the review of our manuscript and we appreciate the suggestions made. In the attached document we provide a response to each review point. The comments are supported with a new version of our manuscript in which we have processed review comments from all reviewers.

Reviewer 3 Report

1.       Reference should add literature from the past 3 years as much as possible;

2.       Table 2 should be further streamlined;

3.       Part 3 Content Analysis, is the core content of the article and lacks sufficient argumentation;

4.       The article needs to further clarify the research focus, controversial points, and research trends of environmental impact factors;

5.       Please pay attention and check if the article is written in accordance with the editorial requirements of the publisher.

Minor editing of English language required.

Author Response

(The authors gave the same response as above.)

Author Response

(The authors gave the same response as above.)

Round 2

Reviewer 1 Report

the authors have carefully addressed the issues, the draft is pbulishable after minor revision.

Some minor issues here:

line33, "2.7GtCO2еquivalent", insert a space between  "2.7Gt" and "CO2еquivalent".

line144, Fig.1 spell out "PRISMA", check all the Figs and Tables for the same issue, make sure that the abstract, all the table and figure shoud be self-understandable, readers can well understand the information without checking textbody.

line429, "net zero", the reviewer guess that next zero CO2 emission. please confirm and revise.

the quality of English is good, only minor editing is needed.

Author Response

Thank you for your useful feedback. We have made the adjustments to the text as suggested. We've changed the last sentence to state : 'on the path to sustainable and low-carbon health systems ' instead of ' on the path to net zero healthcare' to include both carbon emission and other environmental impacts as we have done throughout the research.

Reviewer 3 Report

The authors have carefully responded to the review comments and I believe that this article meets the conditions for publication.

Author Response

Thank you so much for your time and effort to review our paper and advise to publish it in its current form.